# Salivaomic Biomarkers—An Innovative Approach to the Diagnosis, Treatment, and Prognosis of Oral Cancer

**DOI:** 10.3390/biology14070852

**Published:** 2025-07-13

**Authors:** Katarzyna Starska-Kowarska

**Affiliations:** 1Department of Physiology, Pathophysiology and Clinical Immunology, Department of Clinical Physiology, Medical University of Lodz, Żeligowskiego 7/9, 90-752 Lodz, Poland; katarzyna.starska@umed.lodz.pl; Tel.: +48-42-2725237; 2Department of Otorhinolaryngology, EnelMed Center, Drewnowska 58, 91-001 Lodz, Poland

**Keywords:** non-invasive diagnosis, omics biomarkers, oral cancer (OC), oral potentially malignant disorders (OPMDs), saliva, salivaomics, squamous cell oral carcinoma (OSCC)

## Abstract

Salivaomic marker analysis is a revolutionary approach for detecting and planning treatment for oral cancer (OC). Salivaomics assessment, at the gene, protein, and cell levels, is becoming an increasingly common alternative to traditional diagnostic and prognostic methods, representing a non-invasive, efficient, and patient-friendly technique for the recognition of metaplasia and neoplastic changes. It provides a perspective on the carcinogenesis and invasiveness of OC based on genomic, transcriptomic, proteomic, metabolomic, and microbiomic analyses. This review provides a comprehensive overview of the current understanding of the use of salivaomics as a revolutionary method for the early detection of OC and possible treatment modification.

## 1. Introduction

Oral cancer (OC) comprises a group of malignant neoplasms of the head-and-neck region developing from the mucosa of the oral cavity, i.e., lips, tongue, cheeks, floor of the mouth, and hard and soft palates [1,2]. More than 90% of OC cases are squamous cell carcinomas (OSCCs) [3]. According to recent data from Global Cancer Statistics (GLOBOCAN 2020) [GLOBOCAN; https://gco.iarc.fr/today (accessed on 30 May 2025)], OC is the eighth most common cancer in the world, responsible for 377,913 new cases and 177,757 deaths in individuals with OC, accounting for 3% of all cancers and ~1.5% of all cancer deaths. Importantly, the incidence of OC is predicted to increase by 30% by 2030, i.e., with 1.08 million new cases per year [4,5]. According to the latest version of GLOBOCAN 2020 and the International Head and Neck Cancer Epidemiology Consortium, INHANCE, oral cancer is characterized by a higher incidence in men [4,5]. According to epidemiological data, the highest number of newly diagnosed cases and the highest cumulative risk of this type of cancer concern the lips and oral cavity (264,211 new cases per year; age-standardized rate, ASR = 6.0/100,000 population; cumulative risk for age 0–74, CR = 0.68% for men). Similarly, oral cancer shows higher mortality rates in men (125,022 cases per year; ASR = 2.8/100,000 population; cumulative risk for age 0–74, CR = 0.32%). Unfortunately, the development of oral cancer is associated with frequent occurrence of nodal metastases and local recurrences and acquisition of chemoradioresistance; therefore, these tumors are associated with high mortality and morbidity despite significant progress in surgical treatment techniques and the use of increasingly new therapeutic approaches. Importantly, nearly 60–70% of OC cases are diagnosed in the advanced clinical stage of the neoplastic disease, i.e., in the WHO classification stages III and IV; this remains the most common cause of low global overall survival of patients (5-year survival rates still do not exceed 40–60%). Unfortunately, morbidity and survival rates in the case of oral cancer have not improved satisfactorily in recent years [4,5,6,7,8,9].

Oral squamous cell carcinoma (OSCC) is one of the most common head and neck cancers (HNSCCs). The National Comprehensive Cancer Network (NCCN) reports that it accounts for almost 90% of HNCSS cases and is usually associated with poor prognosis, posing a serious challenge to biomedical sciences [10,11,12,13]. It has been etiologically linked with tobacco smoking or chewing betel and alcohol consumption, as well as with exposure to environmental pollutants and other carcinogens. Tobacco smoke tar in particular contains over 7000 toxic substances, with many being carcinogenic. The most important onco-toxins include benzo(a)pyrene, a key polycyclic aromatic hydrocarbon (PAH), as well as nitrosamines (TNA) and N′-nitrosonornicotine (NNN), which increase the risk of oral cancer development, its aggressive invasion, and the occurrence of nodal and distant metastases [14,15,16]. Acetaldehyde, the main metabolite of alcohol, a co-carcinogen, also promotes atrophic changes in the epithelium and facilitates the mutagenic effect of tobacco smoke carcinogens by dissolving them; indeed, alcohol consumption has also been found to increase the risk of OSCC in smokers who are also heavy drinkers [14,15,16]. According to data from the Cancer Genome Atlas Network (TCGA), the development of tobacco-dependent cancers is associated with the occurrence of numerous aberrations in key genes related to carcinogenesis and further tumor invasion. These aberrations include changes in genes regulating the cell cycle (*CDKN2A* and *CCND1*), cell division and its viability (*TP53*, *HRAS*, *PIK3CA,* and *EGFR*), cell differentiation (*NOTCH1*), and mutations increasing the activity of proto-oncogenes, i.e., *c-MYC*, *c-KIT*, *HER-2*, *RAS*, *BCL-2*, and *STAT3*, and inhibiting the functions of anti-oncogenes, i.e., tumor suppressors *RB1*, *p53*, *p16^INK4a^*, and *PTEN* [3,17,18].

An increasingly large subgroup (constituting nearly 38–80% of newly diagnosed HNSCC cases) comprises oropharyngeal cancers (OPSCCs) etiologically related to the presence of human papillomavirus (HPV) infection, mainly HPV-16 and HPV-18, and the integration of HPV genomic DNA into host epithelial cell DNA. Their carcinogenesis is associated with the activity of viral oncoproteins E6 and E7, which contribute to tumor metaplasia and malignant transformation by degrading cell cycle regulatory proteins such as the tumor suppressor p53 [19,20,21,22,23,24,25,26]. Moreover, the viral oncoprotein HPV16/18 E6 affects the function of the oncogene *c-MYC*; this favors the transcription of the human telomerase catalytic subunit (hTERT), resulting in the immortalization of cancer cells and disrupting the activity of CDKs, cyclins, and E2F transcription factors. Furthermore, c-*MYC* can inactivate the function of CDK p27^KIP1^ and p21^CIP1/WAF1^, thereby losing the ability to regulate the cell cycle [27,28,29]. Importantly, HPV-positive cancers have a particular biological profile comprising diagnosis in younger, non-smoking, and non-drinking patients with typically better economic statuses, the achievement of better prognostic parameters, less advanced clinical stages at diagnosis, and lower pT/pN stages; as such, the traditional tumor–node–metastasis (TNM) classification was supplemented with the new AJCC/UICC staging system in 2017. This eighth edition of the American Joint Committee on Cancer (AJCC) recommends the use of a lower-intensity treatment protocol, i.e., de-escalation, for patients with HPV-associated OPSCC [29,30,31].

Most importantly, the gold standard in the assessment of the morphological and clinical stage of oral squamous cell carcinoma, and the response to treatment therapy, patient prognosis, and the risk of OSCC recurrence, is still based around traditional diagnostic techniques, such as aspiration biopsy and surgical biopsy specimens, which are invasive and expensive [32]. These older approaches are hampered by the invasive, time-consuming, and difficult nature of morphological and clinical analysis, and the often delayed diagnosis, resulting in consistently high mortality rates. However, promising non-invasive alternative analytical techniques based around the analysis of saliva and other various body fluids, i.e., plasma, serum, and lymph, have been growing in popularity as they are easy to perform and offer an insight into a range of diagnostic and prognostic biomarkers that have direct contact with OSCC tissues. Moreover, in addition to their ease of collection and the unlimited availability of diagnostic material and ease of transport and storage, only small sample volumes of diagnostic fluid are needed for analysis. Hence, such novel molecular and clinical approaches make saliva analysis (saliva liquid biopsy) a potential sensitive screening tool for detecting early stages of carcinogenesis in the oral cavity; it can be used to determine the advancement of the neoplastic disease stage, predict the response to chemoradiotherapy and drug resistance and the probability of tumor recurrence, and estimate patient prognosis [33,34]. In addition, as saliva contains a number of biomarkers for OSCC, i.e., organic compounds and inorganic components, enzymes, glycoproteins, microRNAs, cfDNA, altered DNA, cytokines, and various other metabolites that are absorbed by the salivary glands from the blood vessels [35], the saliva content constitutes a reflection of the general state of health. Therefore, the current challenge for biologists, pathologists, and clinicians is to identify potential early predictive biomarkers that can be used to improve the early diagnosis of oral cancer, increase patient survival rates, and improve the response to adjuvant therapy S10 = [34].

A liquid biopsy, also known as a liquid biopsy or fluid phase biopsy, is the technique used to sample and analyze non-solid biological tissues such as blood, urine, cerebrospinal fluid, or saliva. Liquid biopsies can detect changes in tumor burden months or years before conventional diagnostic methods. Advances in next-generation sequencing (NGS) or PCR-based methods such as digital PCR are enabling the use of liquid biopsy biomarkers and “omic” parameters as screening tests for many cancers and a method for precision medicine treatment in the future. The liquid biopsy and analysis of saliva, i.e., salivaomics, is a revolutionary method for detecting omics biomolecules in saliva aimed at providing the early diagnosis and accurate prognosis of oral potentially malignant disorders (OPMDs) and squamous cell oral carcinoma (OSCC); it is also a promising research tool aimed at clarifying the pathomechanisms of disease development and invasion [36]. Saliva collection and salivaomics constitute a potentially promising non-invasive alternative tool for making clinical decisions and improving treatment outcomes. In this regard, this tool can be compared to traditional invasive diagnostic methods such as surgical and aspiration tissue biopsy, i.e., the gold diagnostic standard in oral cancer, as well as exfoliative cytology and various imaging methods (positron emission tomography, PET/computed tomography, CT, and fluorescence imaging) used for the early diagnosis of OPMD and OSCC lesions. Salivaomics technology includes detailed analysis of omics markers, i.e., genetic biomarkers (genomics), mRNAs, ncRNAs, and microRNAs (transcriptomics); protein biomarkers (proteomics); epigenetic changes (epigenomics such as changes in DNA methylation, acetylation, ubiquitination, and histone modification at the promoter sites of tumor suppressor genes and oncogenes); metabolic biomarkers (metabolomics); and oral microbiota (metagenomic and microbiomics) (Figure 1) [37].

The growing need to find new, reliable, easy-to-analyze, and low-cost diagnostic omics indicators is related to the constantly increasing morbidity and mortality rates among patients with oral cancer and the lack of specific biomarkers of oral carcinogenesis that would contribute to early detection and progress in therapy. Importantly, despite numerous extensive molecular and histopathological studies and advances in therapeutic methods, no specific and unambiguous prognostic biomarkers for oral cancer have yet been identified. The invasive nature of diagnostic and therapeutic tools is associated with delayed diagnosis, expensive alternative therapeutic methods, and unsatisfactory survival rates in patients with OSCC. Therefore, the search for new sensitive screening methods in oral cancer for the early detection of neoplastic lesions is a serious challenge for biomedical sciences [38,39]. Significant progress has been made in salivaomic biomarker technologies, which has allowed for the detection of both precancerous and neoplastic changes present in the oral cavity; it can also be used for the detection of pathological processes in other tissues based on the evaluation of analytes derived from serum, including proteins, mRNA, and DNA. In this way, salivary biomarkers become diagnostic and prognostic markers in pathological processes beyond the scope of oral diseases. It should be also emphasized that saliva composition varies between individuals, and its nature depends on a range of individual and environmental factors. For example, biomarker content differs between resting and stimulated saliva. Resting saliva is produced by passive collection: the patient tilts the head and allows saliva to accumulate in the mouth without swallowing and then spits it out into the transport unit. Alternatively, saliva production can be stimulated by inter alia paraffin wax or citric acid. In this case, sorbet sets are used, e.g., cotton pads soaked in citric acid, which are placed in the mouth, and saliva is collected using pipettes, pincers, or plastic droppers. In addition to the method of stimulation and collection, the amount and composition of saliva are also influenced by the time of collection, pH, and flow rate. In addition, the concentration and condition of diagnostic markers in the collected saliva are influenced by diet composition, smoking, and alcohol consumption, as well as by the presence of systemic and comorbid diseases, medications, radiotherapy, and condition of the gums and teeth.

Unfortunately, the use of saliva liquid biopsy as a diagnostic and prognostic tool, apart from the previously mentioned advantages, is also associated with several significant limitations that should be remembered when assessing pre- and neoplastic changes (Figure 2) [34,40,41,42].

This narrative review aims to present a comprehensive picture of the latest literature regarding the feasibility and effectiveness of using salivaomic biomarker analysis as a modern, and revolutionary, diagnostic method that can provide an insight into the mechanisms of carcinogenesis initiation; it also examines its potential to identify a more invasive tumor phenotype and determine patient prognosis. Furthermore, the review looks at up-to-date findings concerning the ability of salivaomics to identify the activation of regulatory mechanisms involved in the development of oral cancer, and its drug resistance, contributing to the failure of therapy of oral squamous cell carcinoma. It also discusses the current challenges and future directions of saliva-based omics techniques, highlighting its potential to reduce the risk of carcinogenesis and further development of cancer and improve clinical outcomes in patients with OSCC.

## 2. Materials and Methods

The corpus of research on salivary omics biomarkers included in this review comprised a wide range of recent molecular and clinical studies regarding the use of saliva in the diagnosis, treatment, and prognosis of patients with oral squamous cell carcinoma (OSCC). The work serves as a compendium of knowledge on salivaomics, i.e., the genomics, transcriptomics, proteomics, metabolomics, and microbiomics of saliva, and the clinical use of omics biomarkers; it is based on the latest publications and available data on this topic, selected by the authors of the article.

The following inclusion criteria were used: (a) publications written in English; (b) molecular studies based on human tissues and clinical studies describing the use of saliva-based techniques in the process of diagnosis, treatment, and prognosis of OSCC; (c) cohort studies; (d) retrospective studies; (e) prospective studies; (f) patients with confirmed diagnosis of OSCC. This review also presents the current state of knowledge on the possibilities regarding the pharmacological use of salivaomics in the assessment of resistance to chemoradiotherapy. The final search (conducted 30 May 2025) included the highest-rated peer-reviewed articles published in the past two decades (January 2000 to May 2025), all of which are available through PubMed, Google, Scopus, Cochrane Library, and Web of Science; the articles were also analyzed in accordance with Systematic Reviews and Meta-Analyses Extension for Scoping Reviews (PRISMA 2020) guidelines [43]. The databases were searched using the following keywords: “oral cancer”, “oral squamous cell carcinoma”, “oral potentially malignant disorders”, “OPMD”, “biomarkers”, “omics biomarkers”, “salivaomics”, “saliva”, “diagnosis”, “prognosis”, and “therapeutic or drug resistance”. Additional records were identified using cross-references. A manual search was also conducted combining “AND” or “OR” operators. The paper discusses, in detail, the issues obtained, in order of increasing clinical plausibility.

The following exclusion criteria were applied: (a) unpublished articles or conference proceedings; (b) editorials, opinions, case reports, and letters to the editor; (c) studies in which the patient’s diagnosis was uncertain (e.g., lack of histopathological confirmation); (e) abstracts; (f) articles on using biopsies, blood/plasma samples for OSCC detection.

During data extraction, relevant information regarding the objectives, methods, results, and conclusions was collected from the extracted articles. A total of 401 articles were recognized after the literature search, and of these, 91 pertinent articles were selected for inclusion in the review (Figure 1).

## 3. Studies on the Diagnostic, Prognostic, and Therapeutic Potential of Salivaomic Biomarkers in Oral Cancer

### 3.1. Salivary Genomic Biomarkers for Oral Cancer

#### 3.1.1. DNA Damage and Repair and Salivary DNA Biomarkers

Carcinogenesis in the oral epithelium is associated with the occurrence of numerous mutations and various other disturbances. These include alterations in DNA damage repair mechanisms, DNA deletions or insertions, the multiplexing of chromosomal segments, modified methylation, genetic instability, changes in circulating tumor DNA (ctDNA), single-nucleotide or gene polymorphisms, and epigenetic changes. Many genes and control proteins have been found to play key roles in the development and progression of oral squamous cell carcinoma. Most are caused by progressive genetic changes, such as the inactivation of tumor suppressor genes, such as mutations in p53, and the loss of heterozygosity (LOH) in chromosomes (3p, 9p, 4q, 8p, 11q, 13q, 14q, 17p, and 21p), as well as the activation of oncogenes and microsatellite instability (MSI) due to mutations in DNA mismatch repair (MMR) genes.

DNA can be modified by oxidative stress, i.e., endogenous damage such as in attack by reactive oxygen species produced from normal metabolic byproducts (spontaneous mutation), manifesting as oxidative deamination, replication errors, hydrolysis, and base-pair mismatches. Such changes can also be induced by viruses and ionizing radiation, such as in X-rays, gamma rays, and ultraviolet (UV) radiation, as well as by chemical substances, i.e., mutagenic chemicals, especially aromatic compounds that act as DNA intercalating agents.

DNA damage is recognized by serine/threonine kinase (ATM) and serine/threonine-protein kinase (ATR) enzymes, which also regulate the cell cycle. Impaired ATM/ATR kinase function inhibits the repair function of tumor-suppressor protein p53, a regulatory transcription factor that inhibits p16^INK4a^, cyclin-dependent kinase inhibitor 2A (CDKN2A, multiple tumor suppressor 1), and p52, and regulates the cell cycle, DNA repair, and apoptosis signaling. The levels of the mutated p53 protein have been associated with the early stages of oral cancer. Importantly, antibodies directed against p53 have been detected in the saliva of patients with oral cancer, indicating that saliva testing may be an effective non-invasive method of detecting *p53* mutations [44].

Sreelatha et al. [44] examined the levels of p53 autoantibodies (p53-AAbs) in preoperative saliva samples from patients with histologically confirmed OSCC and in saliva from normal healthy controls who were consuming tobacco. A higher percentage of p53-AAb antibodies was noted in the OSCC patients than in the control group, among whom the results were negative (15% vs. 0). Interestingly, no significant correlation was noted between p53-AAb and the clinical–histological stage (pTNM), although a significant association was found between lymph node involvement and salivary p53-AAb. The mutation of the *p53* TSG gene is one of the most common genetic changes in oral squamous cell carcinomas, which serves as a specific marker of cancer cells in the tissue structure.

Another study assessed the clinical value of hypermethylation status of E3 exon 4 and intron 6 in codon 63 of the p53 gene in saliva from oral squamous cell carcinoma (OSCC) patients [45]. PCR-based analysis indicated a considerably higher frequency of mutations in exon 4 of codon 63 *p53* in the OSCC patients (62.5% of samples) compared to healthy controls (18.52%). In addition, a sufficient amount of good-quality DNA was recovered from saliva to allow PCR amplification, which can be used to detect mutations; the protocol is also fast, cheap, and easy to perform and therefore is suitable for epidemiological studies on oral carcinogenesis.

Mewara et al. [46] also analyzed the C-deletion change of the *p53* gene in exon 4 codon 63 in OSCC saliva samples. Saliva samples were collected from 30 OSCC cases and five healthy controls who did not consume betel nut or smoke tobacco. In the study, p53 gene mutations were confirmed in 93.3% of oral squamous cell carcinoma cases while the result was negative in the control group. The authors confirmed that the detected C-deletion mutation is a potential molecular marker for screening in OSCC patients.

Nandakumar et al. [47] evaluated the significance of salivary 8-hydroxydeoxyguanosine (8-OHdG) expression as a potential marker of the initiation and transformation of premalignant oral disorders (PMODs) such as oral submucous fibrosis (OSMF) to oral squamous cell carcinoma (OSCC). The investigators collected unstimulated saliva from patients with OSCC and OSMF and healthy controls. The obtained data showed that the mean level of 8-OHdG was significantly higher in patients with OSCC than in OSMF cases, with the lowest expression level observed in the control group. The researchers noted that the analysis of OHdG expression provides a non-invasive, simple, and effective way to monitor oxidative stress during oral carcinogenesis and an indicator of DNA damage in OSCC.

Another interesting study was on the evaluation of the expression of eight biomarkers of a saliva supernatant related to oxidative stress, DNA repair, carcinogenesis, metastasis, and cellular proliferation and death in patients with OSCC [48]. The obtained data confirmed a significant increase in the activity of carbonyls, lactate dehydrogenase, metalloproteinase-9 (MMP-9), Ki67, and cyclin D1 (CycD1) and a decreased level of 8-oxoguanine DNA glycosylase, phosphorylated Src (Phospho-Src), and mammary serine protease inhibitor (Maspin). Moreover, the results proved that the activity of the studied biomarkers, especially carbonyls, CycD1, and Maspin, in saliva may be a good diagnostic tool for diagnosis, prognosis, and postoperative monitoring in OSCC.

Moorthy et al. [49] evaluated *EGFR* oncogene and EGFR protein expression as biomarkers in saliva specimens and buccal cells from oral premalignant changes such as oral submucous fibrosis (OSMF) with dysplasia. The highest EGFR immunoexpression was observed in OSCC and OPMD with dysplasia and the lowest in OSMF and healthy controls. In addition, protein expression increased with the degree of dysplasia and clinical advancement of OSCC. An eighteen-fold increase in *EGFR* gene expression was noted in OSCC and three-fold upregulation in OSMF compared to healthy controls. The findings clearly indicate that saliva and exfoliated buccal cells may constitute a clinically relevant non-invasive tool for the assessment of carcinogenesis in the oral epithelium and can be used to identify high-risk patients with OSMF using EGFR as a biomarker.

#### 3.1.2. Telomere Length and Telomerase Expression in DNA Samples: Microsatellie Instability (MSI) and Loss of Heterozygosity (LOH)

Telomeres, the distal ends of linear eukaryotic chromosomes composed of tandemly repeated 5′-TTAGGG-3′ DNA elements, protect the terminal regions of chromosomal DNA from progressive degradation by oxidative stress and genotoxic agents and ensure the proper function of the DNA repair system. The activity of telomeres has also become a point of interest for researchers assessing their expression in both normal oral epithelium and premalignant oral disorders (PMODs) or oral squamous cell carcinoma (OSSC). For instance, Samadi et al. [50] compared telomere length and telomerase expression in saliva samples at various stages of precancerous lesions and oral cancer using DNA extraction and PCR-based telomeric repeat amplification protocol (TRAP) assay. Significantly higher levels of telomerase activity were noted in OSCC compared to PMODs. The researchers concluded that an increased telomerase expression and telomere length may constitute one of the mechanisms contributing to OSCC progression and may represent an important biomarker for its early detection and prognosis. The same researchers also analyzed telomere length and telomerase activity in saliva samples from oral carcinoma patients [51]. It was found that telomerase, a ribonucleoprotein complex responsible for de novo telomere synthesis and the addition of telomere repeats to existing telomeres, may be a respectable indicator of malignant transformation of oral human cells activated frequently during the late stage of POMD lesions; they also proposed that it may play a crucial role in oral carcinogenesis.

In the study of the role of the telomeric shelterin complex, researchers have shown interesting data on the oncogenic role of telomeres [52]. This complex consists of the telomeric repeat-binding factors 1 and 2 (TRF1 and TRF2), the TRF2-and-TRF1-interacting protein (TIN2), the repressor activator protein 1 (RAP1), the protection of telomeres 1 (POT1), and the POT1 interacting protein (TPP1), responsible for the assembly of linear telomeres into a T-loop, which protects them from processing by DNA repair mechanisms. The authors demonstrated that shelterin proteins, overexpressed in various cancers, including OSSCs, are formed during neoplastic transformation from epithelial cells of the oral mucosa and can both initiate the phenomenon of tumorigenesis in the oral cavity as well as determine tumor size, tumor growth, and resistance to treatment.

Two genetic changes characteristic of human cancers, including oral potentially malignant disorders (OPMDs) and oral squamous cell carcinoma (OSCC), are microsatellite instability (MSI) and the loss of heterogeneity (LOH) [53,54]. Microsatellite instability manifests as a genetic predisposition to mutation resulting from impaired DNA mismatch repair (MMR). In turn, LOH is a type of genetic abnormality in diploid cells in which one copy of an entire gene and its surrounding chromosomal region are lost. In oral squamous cell carcinoma, LOH changes are most common in certain regions of chromosomes 3p, 9p, 17p, and 18q, which are associated with the development of human cancer [53,54].

Indeed, El Naggar et al. observed the highest incidence of LOH in saliva and tumor tissue from patients with OSCC in chromosomes 9p, 3p, and 17p [55]. Additionally, 49% of saliva samples and 86% of tumor samples were characterized by the LOH in at least one of the 25 markers tested, although 51% samples lacked any abnormalities. In saliva, 72.2% of 18 patients with the LOH demonstrated a combination of markers D3S1234, D9S156, and D17S799, and the LOH was noted in 55% of twenty saliva samples with cytological atypia and seven out of seventeen samples (35%) without atypia. Interestingly, the authors demonstrated clonal heterogeneity between saliva and the matching tumor in OSCC; this confirmed the genetic instability of the mucosal field, which was also reflected in the saliva flushing tissue in these patients. Importantly, the LOH occurred with a higher frequency at certain chromosomal loci, which was also associated with smoking and the consumption of strong alcohol, two major exposure factors.

Similar conclusions were presented by Arslan Bozdag et al. [54], who conducted a systematic review of studies analyzing the potential role of MSI and the LOH in oral squamous cell carcinoma and the regulatory molecular mechanisms and therapeutic approaches. The highest incidence of the LOH was observed on chromosomes 9p, 3p, 4q, 8p, 11q, 13q, 17p, and 21q; the most commonly observed LOH in OSCC was located on chromosomal arm 9p21, which includes a tumor suppressor gene encoding the CDKN2A protein. Importantly, the dysfunction of the MMR (DNA mismatch repair) system and hyperactive DNA repair pathways results in high genome instability and MSI and is of significance in therapeutic management. This process generates neoantigens. Therefore, immune checkpoint inhibitors, such as pembrolizumab (target PD-1), MPDL3280A (target PD-L1), and ipilimumab (target CTLA-4), make them more advantageous in such situations and therapeutic procedures [56].

#### 3.1.3. Methylation Changes in the CpG Island in Gene Promoters

In addition to genetic changes, the squamous cell carcinoma of the oral cavity may arise following epigenetic events, i.e., those responsible for regulating gene expression without changing the DNA sequence [57,58]. Epigenetic influences manifest as DNA and histone modifications that are not encoded in the DNA sequence; these can include DNA methylation, histone modification, chromosomal remodeling, and microRNA dysregulation, which play a key role in oral carcinogenesis [59]. In OSCC, the most common change is the methylation of CpG islands in the promoter regions of tumor suppressor genes, leading to transcriptional silencing [60,61]. The most important genes that undergo hypermethylation include *RASSF1A* (Ras association domain-containing protein 1, *RASSF2A* (Ras association domain-containing protein 2), *MGMT* (methylated-DNA-protein-cysteine methyltransferase), *DAPK* (death-associated protein kinase 1), and *FHIT* (bis 5′-adenosyl-triphosphatase, also known as fragile histidine triad protein); as these are characteristic of the early stages of OSCC development, they can be considered potential early diagnostic markers of cancer [62].

However, global hypomethylation leading to the activation of oncogenes in the course of oral carcinogenesis is also important. The first genome-wide DNA methylation study was presented by Rapado-González et al. [61], who identified a group of novel DNA methylation markers specific for tongue cancer in the saliva of OSCC patients: *A2BP1* (Fox-1 homolog A, also known as ataxin 2-binding protein 1), *ANK1* (ankyrin 1), *ALDH1A2* (aldehyde dehydrogenase 1 family, member A2), *GFRA1* (GDNF family receptor alpha-1), *TTYH1* (tweety family member 1), and *PDE4B* (cAMP-specific 3′,5′-cyclic phosphodiesterase 4B).

An interesting comprehensive analysis of genome-wide methylation profiles was also presented by Inchanalkar et al. [63], who highlighted epigenetic changes associated with a potentially malignant oral disorder (PMOD), i.e., leukoplakia and gingivobuccal complex cancers (GBC-OSCC), which could be helpful in identifying high-risk precancerous lesions with a tendency for malignant transformation. The researchers identified 32 genes of prognostic significance based on a comparison of methylation with genomic copy number and transcriptome results. Differentially methylated positions (DMPs) were observed in leukoplakia and GBC-OSCC, with the former having a higher number of hypermethylated DMPs. Furthermore, the researchers also identified 45 hypermethylated promoters shared by PMOD and oral cancer in known tumor suppressor genes: *CDKN1B* (cyclin-dependent kinase inhibitor 1B (p27^KIP1^), *ZFP82* (ZFP82 zinc finger protein), *SHISA3*, (Shisa family member 3), *GPX7* (glutathione peroxidase 7), and *IRF8* (interferon regulatory factor 8). The methylation profiles in leukoplakia and cancer were found to differ from those of normal tissues, and the aberrations increased with disease progression. Moreover, copy number loss in *CASP4* (caspase) and gain in *ISG15* (interferon-stimulated gene 15) were found to be epigenetic prognostic biomarkers associated with longer relapse-free, disease-specific, and overall survival, making it a potential prognostic marker for GBC-OSCC.

It is worth noting that new technologies, such as next-generation sequencing (NGS), allow for the identification of genetic changes characteristic of carcinogenesis at a younger age, or after relatively low exposure to carcinogens, in the case of various malignancies [64]. For instance, an analysis of DNA methylation patterns based on bisulfite genomic DNA sequencing found that the epigenetic deregulation of *PTK6* (tyrosine-protein kinase 6) fell during the development of OSCC in young volunteers, and that could serve as a biomarker for the early detection of OSCC and as a treatment target Hsieh = [65].

Several publications have analyzed the specific epigenetic modifications occurring in oral cancer using saliva as a diagnostic fluid. For instance, the hypermethylation of tumor suppressor genes is often located in the chromosome 9p21 region, which contains a cluster of three genes, *p14^ARF^*, *p15^INK4b^*, and *p16^INK4a^*. Some studies have indicated that the detection of p16 hypermethylation in precancer can predict the risk of malignant transformation and may also be used as a prognostic marker. For example, Kaliyaperumal and Sankarapandian [66] describe an interesting study of p16 hypermethylation in oral submucous fibrosis (OSMF), one of the precancerous conditions in the oral cavity. Saliva samples were collected from 30 patients diagnosed with OSMF and 30 control volunteers. It was found that in OSMF, the hypermethylation status of p16 in buccal cells was very high (93.3%) while only partial methylation was noted in saliva samples (50%) and no hypermethylation was noted in control samples. These findings confirm that sampling the buccal epithelium may be a better method of diagnosing oral lesions than sampling saliva and that hypermethylation in the 9p21 region may be a clue related to the transformation of cells into precancerous oral cells in the epithelium.

Another interesting analysis was in a genome-wide DNA methylation study by Rapado-González et al. [61]. The authors included six individuals with tongue squamous cell carcinoma (OTSCC) in their salivary DNA methylation status study. A total of 25,890 CpGs were found, including 5385 hypermethylated CpGs that were differentially methylated (DMCpG) between OTSCC and adjacent non-cancerous tissue. Importantly, 11 genes, viz. *A2BP1*, *ALDH1A2*, *ANK1*, *TTYH1*, *PDE4B*, *GFRA1*, *ACSS3* (acyl-CoA synthetase short-chain family member 3), *CA3* (carbonic anhydrase III), *GABRB3* (gamma-aminobutyric acid receptor subunit beta-3), *HS3ST1* (heparan sulfate glucosamine 3-O-sulfotransferase 1), and *NDRG2* (protein NDRG2), were methylated in at least two CpGs. The DNA methylation level of cancer-specific genes, i.e., *A2BP1*, *ALDH1A2*, *ANK1*, *GFRA1*, *PDE4B,* and *TTYH1*, was characterized by high diagnostic accuracy (≥0.800) in distinguishing patients from controls, indicating them as markers of the early detection of OTSCC using saliva.

González-Pérez et al. [67] also analyzed the *p16^INK4a^* and *RASSF1A* promoter gene methylation in saliva and its association with OSSC. The authors observed higher proportions of promoter methylation of target genes in the saliva of OSCC patients (n = 43) compared to 40 healthy control volunteers (HC; n = 40). Moreover, saliva samples from patients with oral cancers in advanced clinical stages (stage III/IV), poorly differentiated histologically and with severe cellular atypia, were characterized by a significantly higher ratio of methylated than unmethylated target genes.

Liyanage et al. [68] also described the effect of tumor suppressor gene (TSG) silencing by salivary DNA promoter hypermethylation on carcinogenesis in the oral cavity and oropharyngeal region. The study examined the methylation levels of the promoters of *p16^INK4a^*, *RASSF1A*, *TIMP3*, and *PCQAP/MED15* in salivary DNA from patients with oral cavity cancer (OSCC) and pharyngeal cancer (OPC) with respect to their roles as risk factors for tumorigenesis. It was found that *RASSF1A*, *TIMP3,* and *PCQAP/MED15* (mediator of RNA polymerase II transcription subunit 15) were significantly hypermethylated in both OSCC and OPC, and higher TSG DNA methylation levels were noted when smoking, alcohol consumption, and betel chewing were indicated among the exposure factors. In addition, the panel of discussed markers proved to be highly accurate in the early detection of OSCC and OPC.

Interesting observations were also presented by Demokan et al. [69], who investigated the methylation status of a panel of 10 genes (*KIF1A*, *EDNRB*, *CDH4*, *TERT*, *CD44*, *NISCH*, *PAK3*, *VGF*, *MAL,* and *FKBP4*) as potential markers for head and cancers (HNSCCs) in salivary rinses from 101 patients with HNSCCs. The analysis found the promoter hypermethylation of the two genes *KIF1A* and *EDNRB* to be more common in this group compared to 15 healthy subjects. Hence, it appears that the methylation level of the two genes may be a potential biomarker for identifying neoplastic changes. Additionally, patients with stage I and II HNSCCs demonstrated higher *KIF1A* and *EDNRB* promoter CpG methylation in the salivary rinse samples. Remarkably, methylation level did not appear to be influenced by advanced-stage or early-stage tumors.

Interesting data on tumor suppressor gene promoter hypermethylation in *RASSF1A*, *DAPK1,* and the *p16^INK4a^* CpG gene island were presented by Ovchinnikov et al. [70] in a study on salivary DNA from 143 head-and-neck-cancer patients and 31 healthy non-smoker controls. The results indicated that the method could detect a tumor in HNSCC patients with an overall accuracy of 81% compared to healthy non-smoking individuals. Moreover, the panel of three studied genes could be used to detect the of early stages of HNSCC. The researchers emphasized that promoter methylation of the *RASSF1A*, *DAPK1,* and *p16^INK4a^* MSP panel may be a potentially useful method in detecting hypermethylation events non-invasively in patients with HNSCC.

A similar analysis examined DNA methylation patterns in the mediator complex subunit 15 (*PCQAP/MED15*) gene, encoding a co-factor important for regulating the transcription initiation of various promoters; the study was performed in tumor tissue samples from HNSCC patients who were smokers and in samples from healthy volunteers [71]. Saliva from patients with head and neck cancer not related to HPV infection (HNSCC HPV^−ve^) had significantly more methylated DNA clusters of 5′-CpG compared to healthy controls. Similarly, saliva from patients with HPV-negative HNSCC (HNSCC HPV^−ve^) demonstrated a significantly higher DNA methylation of 3′-CpG clusters compared to controls. Interestingly, the authors did not report any significant differences in DNA cluster methylation between HNSCC HPV^+ve^ saliva and healthy controls. As expected, both the HNSCC HPV^−ve^ and HPV^+ve^ samples demonstrated a higher methylation status compared to healthy controls.

Interestingly, Nagata et al. [72] found eight of thirteen tested tumor-related genes to have higher levels of DNA methylation in oral rinse samples from OSCC patients than controls: *ECAD* (epithelial cadherin), *TMEFF2* (transmembrane protein with an EGF-like and two follistatin-like domains *2*), *RARβ* (retinoic acid receptor beta), *MGMT* (methylated-DNA-protein-cysteine methyltransferase), *FHIT* (bis(5′-adenosyl)-triphosphatase, *WIF-1* (Wnt inhibitory factor 1), *p16^INK4a^*, and *DAPK1*. The gene combination, i.e., of *ECAD*, *TMEFF2*, *RARβ*, and *MGMT,* demonstrated both high sensitivity (97.1%) and specificity (91.7%). The results suggest that detecting the hypermethylation of marker genes in oral rinse samples may be a potential tool for the non-invasive confirmation of oral squamous cell carcinoma.

Similar data were also presented by Carvalho et al. [73], who examined methylation levels and hypermethylation patterns in saliva DNA samples from 61 patients with HNSCCs including OSCC. It was found that 54.1% of patients demonstrated methylation in the promoter region of at least one of the tested suppressor genes, viz. *DAPK*, *DCC*, *MINT-31*, *TIMP-3*, *p16*, *MGMT,* and *CCNA1*. The methylation pattern observed in saliva before treatment was not related to the tumor site or clinical stage; however, these patients demonstrated fewer local tumor relapses and lower overall survival. These findings confirm that the hypermethylation pattern constitutes an independent prognostic factor for local recurrence and overall survival. 

Another study examined the methylation of promoters of p16, death-associated protein kinase (DAPK), and O6-methylguanine-DNA-methyltransferase (MGMT) in OSCC [74]. The frequency of p16 methylation was 43%, DAPK methylation 39.7%, and MGMT methylation 39.8%. Unfortunately, in the clinical analysis, the frequency of methylation did not correlate with the clinical outcomes of the disease.

The selected genetic and epigenetic changes in oral cancer, are presented in Table 1.

### 3.2. Salivary Transcriptomic Biomarkers for Oral Cancer

#### 3.2.1. Salivary mRNA

Transcriptomics is the analysis of profiles of different types of RNA molecules (mRNA and microRNA), whose activity can be compared to identify precancerous conditions of the oral epithelium and oral squamous cell carcinoma. Numerous studies have indicated significant differences in RNA expression between patients with OPMD and OSCC and healthy controls, indicating their potential value as biomarkers for use in the diagnosis of oral mucosa lesions. It is now possible to isolate of a wide range of RNA molecules, including mRNAs encoding various proteins important in carcinogenesis; these can include enzymes (MMP, L-fructose, cathepsin V, LDH, AKR1B10, and kallikrein), glycoproteins (cluster of differentiation 44, CD44 and carcinoembryonic antigen, CEA), cytokines (including IL-1β, IL-1RA, IL-4, IL-6, IL-10, IL-13, IL-17A, IL-17F, TNF-α, IFN-γ, HGF, CRP, and VEGF) and other salivary biomarkers (ANG, NUS1, RCN1, transgelin, FSA, PBSA, KPNA2, LGALS3BP, vitamin C, L-fucose, CYFRA 21-1, β2-microglobulin, and cathepsin B) [75,76,77,78,79,80,81,82,83,84].

The last two decades have seen the publication of a number of clinical studies evaluating the importance of salivary transcriptomic biomarkers as indicators of carcinogenesis occurring in the oral epithelium [75,76,77,78,79,80,81,82,83,84], with some examining the significance of IL-8, SAT1, IL-1β, OAZ1, H3F3A, DUSP1, and S100P mRNA in the saliva of OSCC patients [76,77,80]. For example, Cheng et al. [75] compared the mRNA profiles of saliva samples from OSCC patients and patients with oral lichen planus (OLP), a relatively common chronic inflammatory disease of the oral mucosa, considered to be a premalignant disorder of the oral cavity, as potential markers for the early detection of oral cancer. The saliva supernatant samples were collected and divided into five study groups: newly diagnosed OSCC, OSCC-in-remission, disease-active OLP, disease-inactive OLP, and normal controls. Interestingly, higher salivary levels of OAZ1, S100P, and DUSP1 mRNAs were noted in newly diagnosed OSCC patients compared to other patient groups. However, no significant differences were found in IL-8, IL-1β, H3F3A, and SAT1 mRNAs between newly diagnosed OSCC patients and healthy controls.

Similar observations were noted by Michailidou et al. [76], who analyzed the expression of IL-1β, IL-8, OAZ, and SAT based on extracellular RNA in a group of patients with early-clinical-stage OSCC (T1N0M0/T2N0M0) and in subjects with oral leukoplakia and dysplasia, i.e., with mild dysplasia and severe dysplasia/in situ carcinoma. The combination of these four biomarkers was found to have good predictive value for early tumorigenesis (up to 80%) for patients with oral squamous cell carcinoma, but not in those suffering from oral leukoplakia with dysplasia. Moreover, even the combination of only two biomarkers (SAT + IL-8) was found to have significant predictive ability (up to 75.5%) in the early tumor stages.

A study by Li et al. focused on unstimulated saliva samples collected from patients with primary T1/T2 OSCC and healthy individuals [77]. The study evaluated the predictive power of mRNAs isolated from the saliva supernatant in OSCC using a modern microarray method based on the human genome U133A. The microarray analysis showed that over 1600 genes demonstrate increased expression in patients compared to healthy individuals. Moreover, seven transcripts were indicated, i.e., IL-8, IL-1β, DUSP1, HA3, OAZ1, S100P, and SAT, which showed at least a 3.5-fold increase in the saliva of patients with OSCC.

Similar results were presented by Elashoff et al. [78] in a prevalidation of previously indicated salivary biomarkers for oral cancer detection. It was found that all seven biomarkers were present at higher levels in patients with early oral carcinogenesis compared to a group of healthy individuals. Among these, significant differences were noted for IL-8 and SAT mRNA. The results of the validation indicated their value as indicators distinguishing the patients experiencing carcinogenesis from the control group. Also, Brinkmann et al. [79] examined the mRNA expression of DUSP1, IL8, IL1B, OAZ1, SAT1, and S100P in saliva samples: the expression of four transcriptomes (IL8, IL1B, SAT1, and S100P) was significantly increased in patients with late-clinical-stage OSCC (T3-T4) compared to the control group.

However, these observations have not been confirmed universally. Oh et al. [80] compared the expression of potential mRNA biomarkers in OSCC patients and control volunteers as a potential tool for early detection of oral cancer. The results indicated that five transcripts, viz. NAB2, CYP27A1, NPIPB4, MAOB, and SIAE, demonstrated lower expression for the OSCC group compared with a non-tumor group. The data also indicated that the combination of the two mRNAs CYP27A1 + SIAE and lower expression of MAOB-NAB2 mRNAs were of value in the diagnosis of early stages of OSCC, especially in patients under 60 years of age.

Data also indicates that NUS1, RCN1, endothelin-1, ANG, transgelin, FSA, PBSA, KPNA2, LGALS3BP, vitamin C, L-fucose, CYFRA 21-1, β2-microglobulin, and cathepsin B mRNAs also demonstrate variable expression in the saliva of patients with oral cancer. A microarray study by Ueda et al. [81] found the combination of NUS1 and RCN1 mRNAs to accurately distinguish OSCC patients from controls and indicated that this combination can be implemented as a screening test for OSCC; the lower expression of NAB2, CYP27A1, NPIPB4, MAOB, and SIAE was noted for the OSCC group compared to the non-tumor group.

Pickering et al. [82] compared salivary endothelin-1 (ET-1) expression in patients diagnosed with oral squamous cell carcinoma (SCC) prior to treatment with that in healthy volunteers. It was found that 80% of the oral SCC group demonstrated elevated salivary ET-1 levels compared to normal oral epithelium controls. Hence, salivary ET-1 analysis may have value for monitoring patients at risk for oral SCC.

Another recent study by Bu et al. [83] evaluated the levels of salivary transgelin mRNA, a protoncogene biomarker, as a predictor of poor prognosis in patients with OSCC. It was found that transgelin expression was elevated in the OSCC patients compared with normal controls and that this expression was closely associated with various clinical parameters including the T stage, N stage, TNM stage, and differentiation. Importantly, a higher mRNA level of salivary transgelin was associated with a poorer overall survival rate. Interestingly, Malhotra et al. [84] report a 2.75-fold increase in the expression of CK19 mRNA in salivary samples from OSCC patients compared to controls.

#### 3.2.2. Salivary MicroRNA

MicroRNAs (miRNAs) are small regulatory conserved, single-stranded, non-coding RNA molecules containing 21–23 nucleotides. All miRNAs complement a part of one or more messenger RNAs (mRNAs). Nearly 1900 human miRNAs are believed to play a role in RNA silencing and post-transcriptional regulation and are thought to influence the expression of about 60% of human genes. As such, it is likely that miRNAs influence essentially all developmental process and diseases, including carcinogenesis and the development of cancer invasiveness [85]. It has been demonstrated that numerous miRNAs can directly inhibit the genes regulating the cell cycle, thereby controlling cell proliferation, division, apoptosis, and immune responses, as well as intercellular communication and further tumor invasion. Furthermore, miRNA transcripts can regulate carcinogenesis and epithelial mesenchymal transition (EMT) by influencing tumor suppressors or oncogenes. Hence, an analysis of the miRNAs present in diagnostic body fluids, including saliva, may enable the diagnosis and screening of precancerous changes and cancers and facilitate more accurate prognoses; it can also lead to the use of novel cancer treatment strategies based on inhibiting tumor cell proliferation by repairing the damaged miRNA pathway [57].

Many studies have evaluated the clinical potential of the salivary expression of individual miRNAs, or a panel, in oral potentially malignant disorders (OPMDs) or head and neck cancers (HNCs) [85,86,87,88,89,90,91,92]. It was found in a study that microRNAs demonstrate high stability in whole saliva or salivary supernatants and that these biomolecules may be promising reliable diagnostic indicators for oral cancer in high-risk populations such as in tobacco smokers [86]. Modern miRNA analysis methods, e.g., quantitative real-time PCR, microarray, and next-generation sequencing, confirm that miRNA patterns in saliva not only allow for differentiation between the population of patients with oral cancer and healthy controls but offer promise as prognostic indicators for OSCC.

In Momen-Heravi et al.’s [87] study of genome-wide miRNA level patterns in saliva from individuals with oral squamous cell carcinoma (OSCC), several miRNAs were identified. These molecules were significantly deregulated during oral carcinogenesis compared to healthy individuals: eleven miRNAs were underexpressed (miRNA-136, miRNA-147, miRNA-1250, miRNA-148a, miRNA-632, miRNA-646, miRNA668, miRNA-877, miRNA-503, miRNA-220a, and miR-NA-323-5p), and two were overexpressed (miRNA-24 and miRNA-27b). Importantly, a higher expression of miRNA-27b was observed in individuals with OSCC in remission (OSCC-R), oral lichen planus (OLP), and healthy controls.

Similar conclusions were reported by Yang et al. [88], who analyzed the expression of selected miRNAs in individuals with oral lichen planus (OLP) and non-progressive and progressive leukoplakias with low-grade dysplasia (LGD) and in a group of healthy cohort. In OLP, the overexpression of miR-145-5p, miR-99b-5p, miR-181c, and miR-197-3p was demonstrated, whereas in progressive LGD, individuals significantly overexpressed miR-10b, miR-660, miR-708, and miR-30e.

Zahran et al. [89] also investigated the use of salivary microRNAs as potential biomarkers of neoplastic transformation in oral mucosal lesions. The study confirmed a highly significant increase in salivary miRNA-21 and miRNA-184 in OSCC and OPMD lesions (with and without dysplasia) compared with healthy controls. These results clearly confirmed that the assessment of salivary miRNA signature represents a promising tool for monitoring precursor neoplastic lesions and the early detection of disease progression.

Mehdipour et al. [90] also confirmed, in their study of miRNA levels in OSCC and oral lichen planus (OLP), that miR-21 expression was higher in saliva samples from patients with OLP, dysplastic OLP, and OSCC compared to samples from healthy controls. Conversely, reduced levels of miR-125a were observed in OLP, dysplastic OLP, and OSCC samples compared to healthy controls. Furthermore, it was noted that increased levels of miR-31 occurred in patients with dysplastic OLP and OSCC whereas the decreased expression of miR-200a was noted only in the OSCC group. Moreover, it was shown that increased levels of miR-21 in combination with decreased levels of miR-125a in the saliva of patients with OLP may be an indicator of poor prognosis. Conversely, the lack of significant changes in miR-31 and miR-200a levels in saliva of patients with OLP may indicate the absence of malignant transformation.

Similarly Shahidi et al. [91] conclude that salivary miRNA-320a may be a predictive biomarker. The authors noted a significant decline in salivary microRNA-320a in dysplastic OLP and OSCC but not in OLP without dysplasia, indicating that salivary microRNA-320a may have a convenient non-invasive predictive value for dysplastic OLP.

Interesting research results were also obtained in an analysis of miRNA expression in patients who were newly diagnosed as OPMD and healthy controls [92]. The OPMD group were divided into subgroups with four different types of lesions: including leukoplakia, oral submucous fibrosis (OSMF), oral lichen planus (OLP), and OSMF with leukoplakia. It was found that salivary miRNA-21 and miRNA-31 were upregulated in severe dysplasia compared with controls. Among the different lesions, both molecules were significantly upregulated in leukoplakia.

Most studies in this area have focused on the difference between saliva samples from healthy donors and those from HNSCC patients [93,94,95,96]. Salazar et al. [93] identified a panel of microRNAs typical for carcinogenesis in the epithelium of the head and neck organs. A diagnostic panel of three miRNAs, viz. miR-9, miR-134, and miR-191, was differentially expressed between saliva from HNSCC patients and saliva from healthy controls. Of these, miR-127 and miR-191 were upregulated in HNSCC compared to controls while miR-134 was downregulated.

Fahdi et al. [94] report that two salivary miRNAs, miR-let-7a-5p and miR-3928, were downregulated in head and neck squamous cell carcinoma compared to healthy controls; the downregulations also correlated with lymph node metastasis and tumor size, respectively. Salazar-Ruales et al. [95] found the expression of selected microRNAs to be a promising molecular tool for the early diagnosis of HNSCC: significant differences in the levels of miR-122-5p were present between oral cavity cancer and oropharyngeal cancer, miR-124-3p between the larynx and pharynx, and miR-146a-5p between the larynx, oropharynx, and oral cavity. Moreover, miR-122-5p, miR-124-3p, miR-205-5p, and miR-146a-5p proved to be useful tools for differentiating between HPV+ and HPV- OSCC.

Interestingly, Romani et al. [96] showed that among 25 differentially expressed microRNAs, salivary miR-423-5p constitutes a promising diagnostic and prognostic biomarker in oral squamous cell carcinoma. In addition, miR-423-5p appears to be an indicator of the presence of neoplastic changes in patients with OSCC, and higher levels of miR-106b-5p, miR-423-5p, and miR-193b-3p in saliva are associated with shorter disease-free survival (DFS). The researchers believe these findings could be particularly promising for screening studies to monitor high-risk populations and for use in preoperative prognostic assessment if confirmed in an independent cohort of patients. 

The selected transcriptomic changes in oral cancer, are presented in Table 2.

### 3.3. The Salivary Proteomic Biomarkers for Oral Cancer

Numerous studies indicate that the proteomic analysis of saliva may be a promising method for identifying new biomarkers helpful in recognizing changes in the early stages of carcinogenesis in oral cancer and may allow prognosis in cancer patients [100,101,102,103,104,105,106,107,108,109,110,111,112,113,114,115,116,117,118]. An interesting group of proteins circulating in serum and present in saliva are cytokines; these can be pro-inflammatory, e.g., IL-1β, IL-6, and TNF-α; anti-inflammatory, e.g., IL-1, IL-4, and IL-10; pro-oncogenic, e.g., IL-10, TGF-β, LDH, and MMP-9; and pro-angiogenic, e.g., IL-6, IL-8, and VEGF. These regulate the immune response associated with the presence of tumor antigens, as well as cell proliferation and growth in the oral cavity nociceptive epithelium, and angiogenesis and tumor spread.

Zielińska et al. [100] examined the levels of salivary cytokines, viz. IL-17A, IL-17E/IL-25, IL-17F, and TNF-α, and their association with disease advancement in patients with oral and oropharyngeal cancer. The results indicate that salivary concentration was related to the disease clinical stage, with higher values noted in stage III/IV; these increases were also associated with other individual pathomorphological parameters such as T, N, and M.

Another interesting study analyzed the levels of IL-1β, IL-8, and LGALS3BP in early- (TNM stage I-II) and late-stage OSCC (TNM stage III-IV) in 30 PMOD and 29 post-treatment patients [101]. It was found that the concentrations of IL-1β and IL-8 in unstimulated saliva samples were strong discriminators of late-stage OSCC (stage III-IV). LGALS3BP was not found to be increased in late-stage OSCC patients, but was a good discriminator of early-stage OSCC (stage I-II). The studied biomarkers may serve as a potential screening tool, allowing the indication of cancers with a higher stage of advancement before treatment.

Deepthi et al. examined the activity of TNF-α with regard to histologic grading of OSCC and dysplasia in oral leukoplakia and hyperkeratosis [102]. They reported higher concentrations of salivary TNF-α in OSCC compared to leukoplakia and healthy control volunteers. Similarly, a study of the concentrations of selected cytokines (IL-1α, IL-6, IL-8, IP-10, MCP-1, TNF-α, HCC-1, and PF-4) in saliva from OSCC patients, oral leukoplakia (OL) patients, and healthy individuals by Dicova et al. [103] found IL-6 and TNF-α concentrations to be higher in advanced oral cancer compared to early stages. Moreover, neck metastases were associated with increased IL-6 and TNF-α levels. Hence, salivary IL-6, IL-8, TNF-α, HCC-1, and PF-4 may be potential discriminatory factors between OSCC, OL, and healthy controls, and cytokine IL-6 and TNF-α levels may indicate OSCC progression and the presence of nodal metastases.

The level of TNF-α and IL-6 in salivary supernatant was also assessed by Sami et al. [104] in patients with oral cancer, oral benign fibro-osseous tumors (OB-FOL), and healthy controls. The authors also confirmed higher concentrations of both TNF-α and IL-6 in OB-FOL compared to controls, indicating that they may be suitable markers for distinguishing OPMD from neoplastic changes in the oral cavity.

Salivary analytes may also be used as biomarkers for discriminating between potentially malignant oral disorders, oral leukoplakia, and oral cancer. Two works have evaluated the activity of heat shock proteins (Hsps) as indicators of a precancerous state and tumorigenesis in the epithelium of the mammary cavity [105,106]. Bhavana et al. [105] report elevated Hsps27 activity in oral leukoplakia compared with healthy mucosa. Similarly, Bhat et al. [106] note an elevated level of Hsps90 and L-Fucose in saliva in subjects with OPMD and OSCC compared with healthy subjects (both, *p* < 0.05). Hence, the assessment of various cytokines may be a helpful tool in differentiating epithelial hyperplasia from carcinoma in situ and full-blown oral cancer. Indeed, Sharma et al. [107] report a higher level of salivary L-fucose activity in oral potentially malignant disorders (OPMDs) and oral squamous cell cancer (OSCC) and a higher risk of precancerous and neoplastic changes compared to controls.

Other regulatory proteins and metabolites that may serve as potential diagnostic salivary biomarkers include angiotensin (ANG), ANG2, nucleolar complex protein 1 (NUS1), transgelin, reticulocalbin 1 (RCN1), karyopherin alpha 2 (KPNA2), galactin-binding protein-3 (LGALS3BP), cytokeratin fragment 21-1 (CYFRA 21-1), β2-microglobulin, cathepsin B, fibrin A cleavage products (FSA), and paraben–paraben–sulfonic acid (PBSA) [108,109,110,111,112,113,114,115,116,117,118].

Rathore et al. [108] evaluated the expression of cytokeratin fragment 21-1 (CYFRA21-1), a constituent of the intermediate filament protein, in salivary samples from patients with oral potentially malignant disorders (OPMDs) or with oral squamous cell carcinoma (OSCC). Higher expression was noted in patients with oral cancer compared to those with precancerous changes.

Another interesting work concerned the saliva level of mucin (MUC1), a protein protecting the epithelium that plays a role in cell signaling [109]. The highest MUC1 level was noted in patients with OSCC followed by those with premalignant diseases (OPDM) and controls. A similar analysis was undertaken by Azeem et al. [110], who assessed the levels of salivary protein-bound SA (PBSA) and salivary-free SA (FSA) in salivary supernatants; the samples were taken from tobacco chewers with OSCC, tobacco chewers with no premalignant lesions of the oral cavity, and healthy controls. Since PBSA and FSA were elevated in both groups of tobacco chewers compared to controls, the authors indicate that tobacco use may be a crucial confounding factor in the assessment of changes in the oral epithelium.

Other studies have examined the presence of enzymes in the saliva of OSCC patients: MMPs, LDH, AKR1B10, L-fucose, cathepsin V and A, and disintegrin and kallikrein 5 [112,113,114,115,116,117,118]. For example a recent study by Cai et al. [111] analyzed the expression of various angiogenic and oncogenic proteins (ANG, bFGF, EGF, HGF, HB-EGF, VEGF, PDGF-BB, and leptin), and metalloprotease enzymes (MMPs), viz. MMP-1, MMP-2, MMP-3, MMP-8, MMP-9, MMP-10, MMP-13, TIMP-1, TIMP-2, and TIMP-4, in salivary supernatants from OSCC patients and healthy controls. The levels of HGF, PIGF, and VEGF were 2.00-, 8.10, and 1.38-fold higher in the OSCC patients compared to controls. Interestingly, PDGF-BB was identified in 62.5% of OSCC patients, but not in the control group. No significant intergroup difference was observed for the remaining biomarkers (ANG, ANG-2, bFGF, HB-EGF, and leptin). In addition, bioinformatic analysis identified a higher five-year survival rate in OSCC patients with higher expressions of HB-EGF and TIMP-1 compared with a group with lower biomarker levels. These factors correlated with 5-year and cancer-functional states, indicating that they offer promise as saliva-based non-invasive diagnostic/prognostic markers and therapeutic targets for OSCC.

Other studies have examined the presence in saliva of enzymes known to influence the progression of neoplastic changes in the oral epithelium; these include tissue metalloprotease MMPs and their inhibitors, i.e., tissue inhibitors of metalloproteases (TIMPs). Radulescu et al. [112] measured the concentration of MMP-9 and TIMP-2 in a group of 30 patients with OSCC in relation to the Ki-67 proliferative index. The level of MMP-9 positively correlated with the Ki-67 protein and negatively correlated with the levels of TIMP-2. In addition, Chang et al. [113] compared the levels of MMP-1 in saliva samples from healthy individuals (HC), OPMD subjects, and OSCC patients. They observed a higher expression of MMP-1 in the OSCC group compared to the non-cancerous HC and OPMD groups. In addition, in OSCC cases, raised salivary MMP-1 levels were related to a higher pT status, pN status, and histological grade. Also, Smriti et al. [114] compared the salivary MMP-9 concentration between OSCC patients, OPMD subjects, tobacco users, and a control group. Higher MMP-9 levels were observed in the OSCC and OPMD groups than the subjects with tobacco habits and controls. Also, salivary MMP-9 was higher in the poorly differentiated OSCC group than in the moderate and well-differentiated OSCC groups. Taken together, these findings confirm that the activity of enzymes involved in the invasiveness of neoplastic changes and their inhibitors in saliva may be a useful, non-invasive tool in the diagnosis, treatment, and follow-up of OSCC and OPMD.

Other authors have estimated other enzymes as markers of premalignant and cancerous lesions in the oral epithelium [115,116,117,118]. Gholizadeh et al. [115] evaluated the role of salivary LDH enzyme activity in patients with oral squamous cell carcinoma (OSCC), premalignant oral lesions such as oral lichen planus (OLP), and oral lichenoid reactions (OLRs). They found that salivary LDH activity was increased in patients with OSCC compared to other groups. Similar comments were made by Mantri et al. [116], who demonstrated higher LDH levels in saliva of patients with tobacco pouch keratosis compared to those with oral submucous fibrosis (OSMF) or OSCC.

Furthermore, Ko et al. [117] reported a higher activity of aldo-ketoreductase family 1 B10 (AKR1B10) in saliva from patients with oral squamous cell carcinoma (OSCC) compared with healthy controls. Furthermore, high levels of AKR1B10 in saliva were associated with larger tumor sizes, more advanced clinical stages, and the habit of chewing areca. Moreover, OSCC patients with higher levels of AKR1B10 in saliva showed poorer prognosis. Therefore, the results of the study confirmed that AKR1B10 concentration in saliva may be a useful screening biomarker for high-risk patients with OSCC and may be used to monitor the progression of OSCC.

Potential tumor markers in saliva are also proteases, i.e., a group of enzymes involved in a number of physiological and pathological processes, such as the growth, apoptosis, and metastasis of cancer cells. An example was in the study conducted by Feng et al. [118], who assessed the levels of various proteases: matrix metalloproteinases MMP-1, MMP-2, MMP-10, and MMP-12, disintegrin and metalloprotease A (ADAM)9, disintegrin and metalloprotease A with thrombospondin type 13 motifs (ADAMST13), cathepsin V, kallikrein 5, urokinase plasminogen activator (uPA)/urokinase, and kallikrein 7. The authors showed that all these biomarkers had significantly higher concentrations in the saliva of OSCC patients compared to patients with CPD or JBO. Moreover, the use of a three-protease cluster, viz. cathepsin V/kallikrein5/ADAM9, provided an innovative and inexpensive tool for assessing oral health status and for the screening and diagnosis of OSCC.

The selected proteomic changes in oral cancer, are presented in Table 3.

### 3.4. The Salivary Exosomes

Exosomes are membrane-bound extracellular vesicles (EVs) derived from the endosomal pathway as mature multivesicular bodies (MVBs). They are characterized by a double lipid layer and are typically 30–150 nm in size; they are present in the extracellular space of the human body and in fluids such as blood, urine, and saliva [120]. These structures have numerous physiological functions and participate in intercellular communication and intracellular and intercellular signaling, where they act as both biomarkers and potential therapeutic targets. Salivary exosomes from head and neck cancers, including oral squamous cell carcinoma, contain a range of nucleic acids (such as DNA, mRNA, miRNA, and long noncoding RNAs), proteins, and metabolites such as adhesion molecules, cytoskeletons, cytokines, ribosomal proteins, growth factors, oncoproteins, immunomodulatory molecules, and metabolic enzymes, as well as lipids such as cholesterol, lipid rafts, sphingolipids, cholesterol, glycerophospholipids, and ceramides [121,122]. Interestingly, studies have shown that salivary EVs in patients with oral cancer are larger, more numerous, and more diverse than in healthy controls, and their contents are protected from cellular nucleases [121,123].

Most importantly, tumor-derived exosomes (TDEs) may participate in all stages of carcinogenesis and tumor invasion, where they can regulate the initiation, progression, and response to cancer treatment by mediating intercellular communication and intracellular signaling. Such exosomes have a range of bioactive components such as microRNA, transcription factors, and oncogenic and tumor suppressor proteins that can play important roles in tumorigenesis by reprogramming the tumor microenvironment, transferring genetic information, immune tolerance, promotion of metastasis, and bestowing resistance to therapy [122,123,124]. Also, in both the tumor microenvironment and body fluids, TDEs carry functional protein cargoes, RNA cargoes, and dsDNA cargoes, which may serve as potential diagnostic and prognostic biomarkers; such genomic and proteomic payloads also allow TDEs to serve as markers for monitoring cancer progression and therapeutic responses to anticancer therapies [125,126].

Exosomes have been found to occur in different numbers and sizes in the saliva of patients with oral cancer, with interexosomes also being present [127]. A few studies have also indicated that TDE signatures may be good candidates for the early diagnosis, monitoring, and surveillance of pharyngeal and laryngeal cancer associated with HPV-16 infection, the incidence of which is currently increasing, especially among young non-smoking patients [128]. Interestingly, some studies comparing exosomes from HPV^+ve^ and HPV^−ve^ OSCC patients showed that HPV-positive exosomes had low levels of p53 and did not contain cyclin D1, but did contain p16, E6/E7, and the T-cell inhibitory protein PTPN11 [129].

#### 3.4.1. Functions of Exosomes as Potential Biomarkers in Oral Cancer

Exosomes, as bioactive and informational MVB nanoparticles, may influence many aspects of the development and progression of oral cancer. They may modulate the tumor microenvironment, cell-to-cell communication, tumor invasion, and the immune response. Many researchers have demonstrated that liquid biopsies, including those of circulating tumor DNA (ctDNA), circulating tumor cells (CTCs), and exosomal miRNAs present in diagnostic fluids, may have potential clinical applications in head and neck cancers (HNCs), including in oral cancer [130].

Several studies have shown that cancer cells can selectively package selected miRNAs into salivary EVs, which then act as tumor suppressors or oncogenes in head and neck squamous cell carcinoma [131,132,133]. Interestingly, it has been shown that exosomes produced by hypoxic OSCC cells containing microRNAs, i.e., virally delivered mirR-21, induce the epithelial–mesenchymal transition (EMT) of the oral epithelium, leading to enhanced cell migration and invasion and the formation of a pro-metastatic tumor phenotype [134]. Similarly, miR-34a-5p carried by salivary exosomes in OSCC and derived from cancer-associated fibroblasts (CAFs) was found to activate signaling, leading to enhanced tumor proliferation and metastasis via the regulation of the AKT/glycogen synthesis kinase-3β/β-catenin/Snail pathway [135]. Also, salivary extracellular-vesicle-associated miRNAs, i.e., miR-512-3p and miR-412-3p, have been found to as potential biomarkers in oral cancer [136], and exosomal miR-1246 appears to enhance cancer cell motility and invasion by directly activating neoplastic MAPK-activating death-domain-containing 2D in oral SCC [137]. These findings emphasize the potential value of exosomal miRNAs as diagnostic and prognostic biomarkers.

Numerous studies have also highlighted the role of salivary exosomal proteins in regulating intracellular pathways known to promote carcinogenesis and increase the invasiveness of neoplastic lesions [138,139,140,141]. For instance, proteomic analysis found that exosomal proteins MUC5B, HPA, LGALS3BP, A2M, IGHA1, GAPDH, and PKM1/M2 were clearly differentiated in patients with oral cancer compared to healthy individuals; their presence in EVs allowed OSCC to be distinguished from controls with 90% accuracy, indicating that they may represent a promising diagnostic and prognostic biomarker [138]. In addition, exosomal nuclear factor NF-κB-activating kinase-associated protein 1 (NAP1) derived from oral cancer epithelial cell exosomes was found to enhance the cytotoxicity of NK cells by activating the interferon regulatory factor (IRF-3) signaling pathway [139]. Similarly, thrombospondin 1 derived from salivary exosomes of OSCC cells appears to contribute to the polarization of macrophages toward an M1-like tumor-associated phenotype and provoke the extensive invasion of tumor cells [140].

Moreover, exosomes from the saliva of patients with oral SCC promoted tumor growth via the activation of phosphatidylinositol 3-kinase (PI3K)/AKT, mitogen-activated protein kinase (MAPK)/extracellular signal-regulated kinase (ERK), and c-Jun-1 N-terminal kinase/signal transducer and activator of transcription (STAT)-2 pathways [141]. Raulf et al. [142] report that in OSCC patients, the production of exosomes and their presence in diagnostic fluids are modulated by intracellular annexin 1 (ANXA1): a protein that regulates the activity of the epidermal growth factor receptor (EGFR) and alters the release of exosomal phospho-EGFR [142]. Furthermore, exosomal EGFR and phospho-EGFR levels were found to be reduced by cetuximab treatment; this suggests that exosomal EGFR may be a suitable tool for monitoring cetuximab levels [143].

Although these new biomarkers can provide a wide range of possible clinical applications, no validated circulating biomarkers have yet been integrated into clinical practice for head and neck cancer. Moreover, because the isolation of TDEs and their separation are complicated and time-consuming procedures, further additional studies of TDEs as non-invasive biomarkers are needed.

#### 3.4.2. Roles of Exosomes in Regulating Cancer Progression and Metastasis in Oral Cancer

Many studies have analyzed the roles played by the numerous molecules within TDEs in oral cancer initiation and progression. Recent findings clearly indicate that TDEs may support the initiation of tumorigenesis by participating in the transformation of the epithelial to the neoplastic phenotype. They may also facilitate the transformation of the tumor niche toward a metastatic environment by reprogramming non-neoplastic cell functions, thus altering cell–cell communication or autocrine signals. In addition, TDEs may support tumor progression by providing factors necessary to sustain tumor growth via autocrine or juxtacrine signaling [144].

TDEs can also promote EMT by delivering functional complexes by membrane fusion with recipient cells and binding to recipient cell membrane receptors [145]. For example, Kawakubo-Yasukochi et al. [146] showed that miR-200c-3p enhanced the tumor invasive potential by acting in the microenvironment of human oral SCC; also, Li YY et al. [135] observed that EV-secreting CAFs may increase the proliferation and metastasis of oral cancer cells via exosome-mediated paracrine miR-34a-5p signaling.

Furthermore, several recent studies have shown the heterogeneity of TGF-β signaling in oral squamous cell carcinoma. The TGF-β molecule contained in TDEs enhanced the differentiation of fibroblasts into myofibroblasts, thereby promoting tumor progression and metastasis in oral SCC [147]. Interestingly, TDEs in oral cancer can also transfer oncogenic EGFR from tumor cells to endothelial cells, thus increasing the expression of vascular endothelial growth factor (VEGF); this can support invasiveness and metastasis, thus promoting neovascularization and tumor niche formation [148]. Moreover, salivary tumor-derived microvesicles from oral cancer cells can also promote angiogenesis by activating the Shh/RhoA signaling pathway [149]. Furthermore, other investigators have indicated that exosomal miR-150 can promote carcinogenesis by upregulating VEGF expression [150].

Other studies indicate that TDEs can also modulate the host cell immune response in interactions with the tumor microenvironment. For example, Xiao et al. [140] observed that OSCC-derived exosomes have an inductive potential against tumor-associated macrophages (TAMs) and lead to the polarization of these cells from M1 to M2, which promotes tumor metastasis (18). Similar findings were reported by Hsieh et al. [151], who demonstrated that the delivery of miRNA-21-rich exosomes promoted TAM to M2 polarization and EMT. Also, Madeo et al. indicated that EphrinB1-containing exosomes released from the tumor could promote tumor innervation and tumor cell dissemination by inducing axonogenesis [152], and Mutschelknaus et al. [153] found that EVs derived from irradiated HNSCC cells could modify tumor cell motility and promote tumor cell migration via AKT signaling.

Briefly, then, TDEs contain numerous bioactive molecules, including mRNAs, microRNAs, DNA, regulatory proteins, and cytokines, and therefore can alter the function of both epithelial cells and tumor cells and the tumor microenvironment. By actively participating in many aspects of intercellular substance transmission and signal transfer, TDEs can drive tumor initiation, progression and metastasis, resistance to therapy, and immunosuppression. TDEs may therefore serve as potential clinical biomarkers of progression or response to therapy. However, exosomes have not yet been used in the treatment of HNSCC. Further studies are needed to elucidate the molecular mechanisms involved in exosome release and to explore the clinical applications of these vesicles.

### 3.5. The Salivary Metabolomic Biomarkers for Oral Cancer

Metabolomics is another modern and advanced “omics technique” that allows for the recognition of different patterns of metabolites produced in the course of dysplastic changes, precancerous conditions, and oral cancer [154]. The metabolomic analysis of body fluids, including saliva, and the use of modern diagnostic technologies, i.e., time-of-flight mass spectrometry (TOF-MS), capillary electrophoresis, and nuclear magnetic resonance (NMR), enable the indication of intermediate specific products of cellular and tissue metabolism [155]. The diversity of these products characteristic for different stages of carcinogenesis may be a potential tool for the assessment of metabolomic biomarkers and their impact on the detection and progression of OSCC. In the literature on the discussed topic, one can find a considerable number of publications in which the differentiation of metabolic products in oral cancer was made in relation to groups of healthy controls [all below]. An example of such a study was in the publication by Wei et al. [156], who explored the metabolomics of saliva as a diagnostic fluid for stratifying oral cancer and oral leukoplakia (OLK). The use of ultra-performance liquid chromatography coupled with quadrupole/time-of-flight mass spectrometry in the study revealed characteristic metabolic signatures of saliva for these pathological lesions in the oral cavity. The researchers indicated a clear predictive power of a panel of five salivary metabolites, i.e., γ-aminobutyric acid, phenylalanine, valine, n-eicosanoic acid, and lactic acid. In particular, the panel of three metabolites, i.e., valine, lactic acid, and phenylalanine, gave high accuracy, sensitivity, specificity (reaching over 80%), and a positive predictive value in distinguishing OSCC from controls or OLK. Interestingly, the comparison of the metabolic profile of OSCC with OLK and control group revealed higher salivary n-eicosanoic acid and lactic acid levels and lower GABA, phenylalanine, and valine levels [156]. Also Lohavanichbutr et al. [157] found that four salivary metabolites, namely proline, glycine, citrulline, and ornithine, were significantly altered in the OSCC groups. Another interesting study analyzing saliva for specific metabolites was the work by Sugimoto et al. [158]. The authors performed a comprehensive metabolite analysis of saliva samples from individuals with OSCC and periodontal disease and from healthy controls using capillary electrophoresis-time-of-flight mass spectrometry (CE-TOF-MS). Among the 57 major metabolites, small but significant differences were found between the studied patient groups in the levels of products such as phyroline, hydroxycarboxylic acid, choline, tryptophan, threonine, carnitine alpha-aminobutyric acid, and phenylalanine. Similar observations were shown by Wang et al. [159] using ultra-high-performance liquid chromatography–mass spectrometry (UPLC-MS) in hydrophilic interaction chromatography mode. The results of the analysis indicated significantly increased betaine, pipecolinic acid, and choline levels while L-carnitine was decreased in stage I-II OSCC patients compared to healthy individuals. Another study performed by the same group of researchers used hydrophilic interaction chromatography for the analysis of metabolomic biomarkers of OSCC to detect salivary metabolites, including propionylcholine, S-carboxymethyl-L-cysteine, phytosphingosine, sphinganine, and N-acetyl-L-phenylalanine [160]. The authors indicated that the combination of these metabolites provided a significantly satisfactory result in terms of accuracy, sensitivity, and specificity in distinguishing the early stage of OSCC from the controls. Also, Sridharan et al. [161] reviewed the metabolomic profile of saliva from patients diagnosed with oral leukoplakia and oral squamous cell carcinoma and compared the results with the control group using Q-TOF-mass spectrometry. The researchers noted significant increases in the concentrations of 1-methylhistidine, inositol 1,3,4-triphosphate, d-glycerophosphate-2-phosphate, 4-nitroquinoline 1-oxide, 2-oxoarginine, norcocaine nitroxide, sphinganine 1-phosphate, and pseudouridine in oral leukoplakia and OSCC compared to controls. Compounds with downregulation in the patient groups included l-homocysteic acid, ubiquinone, neuraminic acid, and estradiol valerate. In summary, it is worth noting that the analysis of salivary metabolome in cases of oral cancer and precancerous lesions of the oral mucosa may be a valuable and useful diagnostic tool, and the metabolomic approach complements the clinical detection of OSCC and stratifies the two types of lesions, which leads to better disease diagnosis and prognosis.

### 3.6. The Salivary Metagenomic Biomarkers for Oral Cancer

Saliva metagenomics, which examines the genome of numerous oral microorganisms, is an important tool that allows for confirming the presence of many species of bacteria that may affect the occurrence of metaplasia and subsequent carcinogenesis. The diversity of oral microorganisms, characteristic of various stages of neoplastic disease, may also have a potential for prognostication, diagnosis, and progression. Few publications clearly indicate the participation of over 700 bacterial species, assigned to 12 phyla and 185 genera. The latest technologies for studying oral microbiota, i.e., culture, microscopy, DNA microarrays, PCR, 16S rRNA, and sequencing, are used for the precise qualitative and quantitative identification, classification, and study of bacteria. Importantly, salivary dynamics and the temporally and spatially diverse pattern of the microbial community are influenced by different factors such as the clinical stage of advancement of oral lesions, changes in salivary pH, smoking, alcohol consumption, pathogen infection, dental caries and oral and periodontal diseases, and genetic mutations [57,162,163]. Studies have indicated that 70% of culturable species are bacteria, i.e., Firmicutes, Proteobacteria, Fusobacteria, Chlamydiae, Bacteroidetes, Actinobacteria, Spirochaetes, SR1, Chloroflexi, Synergistetes, Gracilibacteria, and Saccharibacteria [57,162,163]. However, the comprehensive profile of the oral microbiome during cancer progression from the early to the late stage is still unclear.

An important aspect of salivary metagenomics analysis is the finding of significant differences in oral microbiota, the symbiotic relationships of complex microbial communities that differ significantly between patients with precancerous conditions and oral cancer and healthy control individuals. An example was in the study by Lee et al. [164], who analyzed the differences in the microbiota between healthy volunteers and patients with epithelial precursor lesions and cancer patients with different lifestyle habits such as betel chewing and smoking. The use of next-generation sequencing allowed them to identify five types of microorganisms typical of the bacterial community composition in OSCC patients, i.e., Bacillus, Enterococcus, Parvimonas, Peptostreptococcus, and Slackia. The analysis revealed significant differences between epithelial precursor lesions and cancer patients and correlation with their classification into two clusters. Another interesting example was in the study by Mager et al. [165], who investigated the composition of salivary microbiota as a diagnostic indicator of oral cancer based on a randomized study of cancer-free and oral squamous cell carcinoma subjects. The researchers analyzed the content of 40 common oral bacteria using the modern checkerboard DNA–DNA hybridization method in a group of 45 OSCC subjects and 45 controls matched by age, gender, and smoking history. The results of the analysis allowed them to indicate three groups of bacteria, i.e., *Capnocytophaga gingivalis*, *Prevotella melaninogenica,* and *Streptococcus mitis*, whose abundance was increased in the saliva of people with cancer. Importantly, diagnostic sensitivity and specificity in the matched group were 80% and 82%, respectively. Also, the analysis of the bacteriome and bacterial dysbiosis was undertaken by Belibasakis et al. [166], who, for a better understanding of the concept of microbial dysbiosis, together with the advancement of Illumina high-throughput molecular sequencing, allowed for a more detailed determination of the microbial character in cases of oral lichen planus (OLP), oral cancer, and a healthy epithelium. The researchers confirmed that the most reliable microbiome pattern involving OSCC and controls included bacteria such as *Prevotella*, *Streptococcus*, and *Salmonella*, *Fusobacterium nucleatum*, and *Porphyromonas gingivalis*.

Numerous studies have also focused on the analysis of mechanisms used by bacterial pathogens leading to changes that affect carcinogenesis and the progression of pathological changes in the oral cavity [166,167,168]. An example was in the study of Belibasakis et al. [166], who suggested that two types of bacteria, *Fusobacterium nucleatum* and *Porphyromonas gingivalis*, stimulate the secretion of proinflammatory cytokines that trigger the progression of neoplastic changes by influencing the uncontrolled proliferation and subsequent invasiveness and metastasis of tumor cells. Similar prognostic conclusions were reached by Gopinath et al. [167], who indicated that *Porphyromonas gingivalis* contributes to proinflammatory cytokine TNF-α secretion and metalloproteinase production and inhibits the DNA repair role of protein p53. Moreover, *Fusobacterium nucleatum* also contributed to cancer development through the secretion of cytokines and lipopolysaccharides. Other researchers have also profiled the oral microbiome and identified bacterial biomarkers associated with OSCC. An example was in the study by Yang et al. [168], who analyzed the oral microbiota of 51 healthy controls and 197 patients with OSCC at different stages. The analysis used the 16S rRNA V3V4 amplicon sequencing method, as well as bioinformatic and statistical analysis. Interestingly, the bacterial microbiota populations changed dynamically with the progression of carcinogenesis and tumor invasiveness in clinical stages from SI to SIV. For example, the Fusobacteria biota significantly increased with the progression of OSCC and significantly differed from healthy individuals. On the other hand, the abundance of Streptococcus, Haemophilus, Porphyromonas, and Actinomyces biota decreased with the progression of the neoplastic disease. The researchers also pointed to a panel of bacterial markers of three bacteria: the increased expression of Fusobacterium periodonticum and decreased expression of *Streptococcus mitis* and Paenibacillus pasteuri, which differentiated the neoplastic lesions from the healthy oral epithelium. Interestingly, the authors noted that genes involved in carbohydrate metabolism, such as methane, and parameters of energy metabolism, such as oxidative phosphorylation and carbon fixation in photosynthetic organisms, were upregulated in late-stage OSCC whereas genes involved in amino acid metabolism, such as folate biosynthesis and valine, leucine, and isoleucine biosynthesis, were significantly associated with healthy controls. Schmidt et al. [169] also profiled the bacteria in OSCC tumor and anatomically matched contralateral normal tissues from the same individuals using 16S rDNA hypervariable region amplicon sequencing. The study showed that *Firmicutes* spp. (particularly Streptococcus) and *Actinobacteria* spp. (particularly Rothia) were significantly reduced in tumor samples compared to normal tissue samples from the same patient. Furthermore, the investigators found that *Streptococcus* spp. caused DNA strand damage through the production of acetaldehyde, leading to lipid peroxidation, while *Lactobacillus* spp. contributed to the establishment of a cancer-friendly environmental niche and Acinetobacteria played a key role in the development of periodontal disease and caries. The study of oral microbiota was also undertaken by Thomas et al. [170], who studied the influence of alcohol consumption and tobacco smoking on changes in human oral microbiome biofilms. The researchers used DNA extraction from swabs, and the V1 region of the 16S rRNA gene was amplified by PCR and sequenced using the Ion Torrent PGM platform. Interestingly, the authors also observed differences in species diversity between the groups of smokers and smokers/alcohol drinkers and the control group. A significant decrease in the abundance of Neisseria, Granulicatella, Staphylococcus, Fusobacteria, Peptostreptococcus, and Gemella was characteristic of smokers and smokers/drinkers compared to the control group. However, a significant increase in the levels of Prevotella and Capnocytophaga was observed in both smoking and smoking/drinking OSCC patients. Interestingly, controls showed a higher abundance of Aggregibacter.

In summary, the results of numerous studies on oral biota clearly indicate that different bacterial communities inhabiting the oral cavity change in terms of abundance and activity during the progression of oral cancer. They therefore represent potential and promising diagnostic markers of OSCC and are indicators of the progression of neoplastic lesions of the oral epithelium.

## 4. Discussion

### 4.1. Current Challenges in the Use of Salivaomics in Oral Cancer

The use of salivaomics in the diagnosis and prognosis of cancer presents a number of challenges. Indeed, improving its effectiveness and reliability in detecting and prognosticating oral cancer (OC) is still beset by various difficulties and new approaches are needed to make the widest possible use of the advantages of this modern diagnostic and clinical technology [171,172,173,174]. It is important to achieve consistency, unambiguity, and reliability between research results, and this is best achieved by standardizing saliva collection, storage, and processing techniques. Therefore, efforts to identify the components of this “specific salivary biomolecule circulation system” require thorough and extensive clinical validation, which should take into account the clinical factors and individual variation that can influence the composition of biomarkers and their levels in saliva. These challenges can be solved by advanced, sensitive, and cost-effective technologies and complex data analysis methods that can detect biomarkers even at very low concentrations. Unfortunately, the validation process is long and complex and is associated with the continuous, and tedious, development of new technologies for the identification of salivary biomarkers. Furthermore, further challenges are posed by the need to gain acceptance by diagnosticians, biologists, and clinicians for incorporating salivary diagnostics into routine clinical practice.

Salivaomics has key limitations, viz. (a) potential cross-contamination; (b) the considerable variability of saliva composition, being influenced by inter alia circadian rhythms, food consumption, oral hygiene, and the overall state of health; (c) lower concentrations of biological molecules in saliva compared to other body fluids; (d) the influence of the stimulation method (using paraffin wax or citric acid); and (e) the lack of clarity regarding the transfer of omics components from plasma, serum, or lymph to saliva. As such, the use of saliva as a diagnostic medium and a potential tool for screening and diagnosis requires further validation [175,176,177,178]. The method of saliva collection also has an impact on the results, i.e., the use of passive saliva collection methods or the use of sorbet kits (Salimetrics^®^, Carlsbad, CA, USA contains cotton buds) or an ophthalmic sponge (Merocel^®^, Medtronic, Minneapolis, MN, USA) or a cotton swab (microFLOQ^®^, Copan, Italy), which are placed under the tongue for 30 s to collect saliva [179,180].

Certain guidelines should be followed to ensure accurate and reliable salivaomic data. Firstly, to maintain a constant and comparable composition of saliva and reduce the risk of cross-contamination, samples should be collected in a resting state, typically in the morning, preferably at least two hours before breakfast; however, it is acceptable to collect saliva samples at standardized intervals postprandial for the analysis of metabolites such as glucose and lactate because they are less dependent on circadian rhythms. Secondly, to minimize the risk of degradation of important salivary biomarkers, the diagnostic material should be properly stored, i.e., omics analyses should be performed as soon as possible after collection; the tests can be postponed for several hours to a month after collection by storing the saliva at 4 °C ÷ 6 °C and −20 °C ÷ −80 °C, respectively. If the samples are not centrifuged, they should be analyzed within 24 h; otherwise, if centrifuged at 1500 rpm for two minutes to remove cell debris, the analysis may be postponed. Thirdly, the analysis should employ advanced procedures and sensitive and specific technologies. These can include mass spectrometry, polyacrylamide gel electrophoresis (PAGE), and matrix-assisted laser desorption ionization with time of flight (MALDI-TOF), advanced PCR, and microarray for studying proteins, DNA, and RNA; gas chromatography–mass spectrometry (GC-MS) and next-generation sequencing can be used to analyze microbiome communities [57,179,180,181,182]. Following standardized protocols for saliva collection, processing, and storage improves the standard of the obtained data, helps minimize variability, and ensures the repeatability and comparability of results [183].

### 4.2. Future Directions in the Use of Salivaomics in Oral Cancer

To fully realize the diagnostic and therapeutic possibilities of salivaomics, future directions should take account of the continual progress achieved in the area through the integration of genomics, proteomics, metabolomics, and transcriptomics. The identification and use of new saliva biomarkers requires the constant validation of their application. Also, further large-scale studies are needed with large cohorts of patients using modern methods, such as nanotechnology, microfluidics, and biosensors. The use of such automated analytical systems can minimize human errors and increase the repeatability of results in the studied populations. It is also important that studies on real-time saliva diagnostics take advantage of equipment that is easy to use and interpret, portable, and patient-friendly and is inexpensive. Such approaches can also draw on artificial intelligence (AI), including machine learning (ML) and deep learning (DL) strategies, as modern methods for better understanding and using salivaomics in screening for cancers and other diseases.

Such a multifaceted approach can yield clearer conclusions regarding diagnostics and prognosis. It also offers the possibility of creating a repeatable method of patient stratification, which can be helpful in assessing the risk of disease development and designing individual preventive strategies and personalized treatment. Saliva-based diagnostics can allow the effective early detection of disorders, including malicious transformation, monitoring the response to treatment, and be used to prevent neoplastic relapse; as such, it represents a non-invasive and widely accepted diagnostic tool that offers considerable promise in public health screening programs. However, its widespread adoption requires clear guidelines on the standards of collection, storage, and preparation of saliva samples for analysis and the continuous improvement of the approval process.

## 5. Conclusions

Oral squamous cell cancer (OSCC) is currently the most common form of head and neck cancer. Unfortunately, despite extensive molecular studies and the use of modern microarray methods, OSCC has been characterized for many years by unacceptable high rates of morbidity and mortality. Therefore, there is great interest in identifying new diagnostic methods and biomarkers that would allow for an unambiguous early diagnostic and prognostic method that can determine the most beneficial form of personalized treatment for patients with OSCC.

Unfortunately, the available methods for detecting early epithelial changes typical for oral carcinogenesis are invasive and expensive to perform. Also, the standards for interpreting the obtained data are inconsistent and the area lacks unequivocal diagnostic and prognostic indicators and biomarkers. As such, treatment failures, such as local tumor recurrence, regional and distant metastases, and resistance to chemoradiotherapy, are common, resulting in high morbidity and mortality.

Salivaomic analysis is becoming a potentially valuable diagnostic and prognostic method in oncology. Saliva is an easily accessible diagnostic material that reflects changes in the oral mucosa. It can be collected without invasively affecting the epithelial structure and can be obtained and stored at low cost. However, it currently competes with various other techniques, such as aspiration, fine- and core needle tissue biopsy, surgical biopsy, and exfoliative cytology, as well as various imaging modalities, such as positron emission tomography (PET), computed tomography (CT), and fluorescence imaging.

Saliva, being a filtrate of body fluids such as plasma, serum, and lymph, reflects changes in the molecular profile of the entire organism at the genomic, transcriptomic, proteomic, metabolomic, and metagenomic levels. It can thus provide a wide-spectrum understanding of the molecular landscape of oral cancer, opening the way to identifying potentially critical multiomic biomarkers that may be used to determine the clinical stage of neoplastic disease.

However, to ensure reliability, unambiguity, and repeatability, it is necessary to validate and standardize saliva collection and processing protocols with the aim of eliminating potential cross-contamination and the problems associated with compositional variability. Hence, further research into the multiomic composition of saliva is needed before it can be included in routine clinical practice. Nevertheless, following validation, salivaomics represents an economically viable option for both large-scale screening studies and the design of individual treatment. 

## Data Availability

Data are available in a publicly accessible repository.

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
