# Peer review of "Salivaomic Biomarkers—An Innovative Approach to the Diagnosis, Treatment, and Prognosis of Oral Cancer"

_biology, 2025, doi:10.3390/biology14070852_

Round 1
Reviewer 1 Report
Comments and Suggestions for Authors
I recommend reevaluating the sections where p values are cited, you should remove it
Author Response
Response to Reviewer 1 Comments
I would like to thank you for your considered, substantive and helpful review of my work. Your suggestions have been taken into consideration in my revision, and have made an invaluable contribution to the redrafting and editing of the revised text.
Point 1: I recommend reevaluating the sections where p values are cited, you should remove it.
Response 1: I agree with the Reviewer's #1 comments. The numerical p value has been removed from all sections where it was cited.
Point 2: The English could be improved to more clearly express the research.
Response 2: The manuscript has been proofread by an English native speaker from the University. A certificate from the Foreign Language Centre, Medical University of Lodz, Poland is attached.
I thank the Reviewer for reviewing my work. I hope that the new version complies with the Reviewer’s suggestion, and may be again considered for publication in Biology.

Reviewer 2 Report
Comments and Suggestions for Authors
I appreciate the efforts of authors for their work in presenting a comprehensive overview of using salivary biomarkers for efficient diagnosis, treatment and prognosis of Oral Cancer. Although the review significantly outline the salivary omics biomarkers involved in OSCC based on the available research literature, there is scope to improve the scientific rigor and reader utility.
After going through the manuscript, following revisions are recommended that will surely going to enhance the quality of the review:
- Inclusion of comparisons between markers from whole saliva and other biological sources (serum/plasma, tissue etc.)
- Compare the usage of whole saliva and salivary exosomes/extracellular vesicles in context to Oral Cancer diagnostics. Expand this section to cover validated exosomal biomarkers in Oral Cancer.
- OSCC is the chief form of Oral Cancer, it does not represent all types of Oral Cancers, therefore while using this terminology in “Simple Summary” or in other sections of manuscript, it is required to familiarize the readers with this concept. It is advised to use “OC” abbreviation for Oral Cancer and “OSCC” for Oral squamous cell carcinoma.
- Tables illustrating various biomarkers should be supported with quantitative evidence by adding diagnostic metrices (sensitivity, specificity) and methods (qRT-PCR, LC-MS/MS).
- Term like “liquid biopsy” should be clearly defined at first mention for the benefit of interdisciplinary readers.
Overall, the manuscript is scientifically solid and addresses an important topic, with some focused revisions mentioned above along with a stronger discussion of real-world applications, it will be well-prepared for publication.
The manuscript is generally well-written and readable, with a good command of scientific terminology. However, improvements are needed in conciseness and clarity. Several sections are overly verbose or contain repetitive statements, particularly regarding OSCC diagnostic challenges. A thorough language edit is recommended to polish the manuscript for publication.
Author Response
Response to Reviewer 2 Comments
I would like to thank you for your considered, substantive and helpful review of my work. The current version of the revision addresses as many points as possible raised in the review.
I appreciate the efforts of authors for their work in presenting a comprehensive overview of using salivary biomarkers for efficient diagnosis, treatment and prognosis of Oral Cancer. Although the review significantly outline the salivary omics biomarkers involved in OSCC based on the available research literature, there is scope to improve the scientific rigor and reader utility. After going through the manuscript, following revisions are recommended that will surely going to enhance the quality of the review:
Point 1: Inclusion of comparisons between markers from whole saliva and other biological sources (serum/plasma, tissue etc.)
Response 1: Unfortunately, as the author of the manuscript, I cannot fully agree. The text of the work intentionally focuses exclusively on the analysis of biomarkers in saliva, with the aim of clarifying the state of research regarding salivaomics. As this topic is already highly complex and multidimensional, it was decided to omit a detailed comparative analysis with other liquid markers detected in biological material of other origin, such as tissue, serum or plasma. However, some analyses of biomarkers from material other than saliva were included, and these are clearly indicated in the text. Moreover, the limitations section emphasizes the difficulty of comparing data obtained from different biological materials.
Point 2: Compare the usage of whole saliva and salivary exosomes/extracellular vesicles in context to Oral Cancer diagnostics. Expand this section to cover validated exosomal biomarkers in Oral Cancer.
Response 2: Section 3.4. "The Salivary exosomes" includes a detailed discussion of the usage of salivary exosomes/extracellular vesicles in the context of oral cancer diagnostics and prognosis. These issues are introduced in two subsections: 3.4.1. Functions of Exosomes as Potential Biomarkers in Oral Cancer and 3.4.2. Roles of Exosomes in Regulating Cancer Progression and Metastasis in Oral Cancer. Arguably, a more extensive analysis of publications concerning exosomes present in biological material other than saliva would unnecessarily complicate the discussion of salivary biomarkers present in salivary extracellular vesicles.
Point 3: OSCC is the chief form of Oral Cancer, it does not represent all types of Oral Cancers, therefore while using this terminology in “Simple Summary” or in other sections of manuscript, it is required to familiarize the readers with this concept. It is advised to use “OC” abbreviation for Oral Cancer and “OSCC” for Oral squamous cell carcinoma.
Response 3: I entirely agree. In accordance with Reviewer's #2 suggestion, in the "Simple Summary" section and in the text of the paper, the terminology of the discussed neoplastic disease was unified and the abbreviation "OC" was used for Oral Cancer and "OSCC" for Oral Squamous Cell Carcinoma. The difference in the terms "OC" and "OSCC" was defined in the Introduction section.
Point 4: Tables illustrating various biomarkers should be supported with quantitative evidence by adding diagnostic metrices (sensitivity, specificity) and methods (qRT-PCR, LC-MS/MS).
Response 4: Unfortunately, I cannot fully agree. Any missing data regarding salivaomics biomarkers, information about diagnostic methods and p values ​​used in specific studies has been added to the tables where necessary. However, metrics (sensitivity, specificity) have been intentionally omitted to more clearly present the results of the multivariate and complex analyses, the aim being to present the results of many studies in the most reader-friendly way. Furthermore, including new metric data would not affect the final conclusions or results.
Point 5: Term like “liquid biopsy” should be clearly defined at first mention for the benefit of interdisciplinary readers.
Response 5: I entirely agree. In accordance with the Reviewer's #2 suggestion, the term “liquid biopsy” was clearly defined and included in the text of the Introduction.
…” A liquid biopsy, also known as a liquid biopsy or fluid phase biopsy, is the technique used to sample and analyse non-solid biological tissue, such as blood, urine, cerebrospinal fluid or saliva. Liquid biopsies can detect changes in tumor burden months or years before conventional diagnostic methods. Advances in next-generation sequencing (NGS) or PCR-based methods such as digital PCR are enabling the use of liquid biopsy biomarkers and “omic” parameters as screening tests for many cancers and a method for precision medicine treatment in the future.”
Point 6: The English could be improved to more clearly express the research.
Response 6: The manuscript has been proofread by an English native speaker from the University. The certificate from Foreign Language Centre, Medical University of Lodz, Poland has been attached.
I thank the Reviewer for reviewing my work. I hope that the new version complies with the Reviewer’s suggestions, and may be again considered for publication in Biology.

Reviewer 3 Report
Comments and Suggestions for Authors
The manuscript submitted by Katarzyna Starska-Kowarska describes that Salivaomic Biomarkers – An Innovative Approach to The Diagnosis, Treatment and Prognosis of Oral Cancer. TThis review is of great importance and interest for the future diagnosis of oral cancer. There is nothing wrong with its content. However, the nature of saliva is very unstable; even on the same day it is different in the morning and in the evening. The nature of saliva also differs between resting saliva and stimulated saliva. If we are going to use it for diagnosis, we need to clear up these issues, what do you think? If possible, please include my points somewhere in the text.
Author Response
Response to Reviewer 3 Comments
I would like to thank you for your considered, substantive and helpful review of my work. All your comments and suggestions have been taken into consideration in this revision, and they have made an invaluable contribution to the redrafting and editing of the revised text. A point-by-point answer to the Reviewer’s comments is given below.
Point 1: The manuscript submitted by Katarzyna Starska-Kowarska describes that Salivaomic Biomarkers – An Innovative Approach to The Diagnosis, Treatment and Prognosis of Oral Cancer. This review is of great importance and interest for the future diagnosis of oral cancer. There is nothing wrong with its content. However, the nature of saliva is very unstable; even on the same day it is different in the morning and in the evening. The nature of saliva also differs between resting saliva and stimulated saliva. If we are going to use it for diagnosis, we need to clear up these issues, what do you think? If possible, please include my points somewhere in the text.
Response 1: I entirely agree. In accordance with the Reviewer's #3 suggestion, in the Introduction section, information about the variation in salivary secretion between individuals, and the fact that it is dependent on many individual and environmental factors, was added to the text. It was emphasized that when analyzing the levels of biomarkers contained in saliva, it is necessary to take into account whether it is resting saliva or stimulated saliva. It was also noted that the quantity and composition of saliva depend on the time of collection of the material at different hours of the day, the saliva stimulation method, collection method, pH and flow rate. Furthermore, the revised version indicates that the concentration and condition of diagnostic markers in collected saliva is influenced by the composition of diet, smoking and alcohol consumption, as well as systemic and comorbid diseases, drugs, radiotherapy, condition of gums and teeth. All mentioned factors that influence the quantity and quality of saliva were also presented in Figure 1.
…”It should be emphasized that saliva composition varies between individuals, and its nature depends on a range of individual and environmental factors. For example, biomarker content differs between resting and stimulated saliva. Resting saliva is produced by passive collection: the patient tilts the head and allows saliva to accumulate in the mouth without swallowing, and then spits it out into the transport unit. Alternatively, saliva production can be stimulated by inter alia paraffin wax or citric acid. In this case, sorbet sets are used, e.g. cotton pads soaked in citric acid, which are placed in the mouth and saliva is collected using pipettes, pincers or plastic droppers. In addition to the method of stimulation and collection, the amount and composition of saliva are also influenced by the time of collection, pH and flow rate. In addition, the concentration and condition of diagnostic markers in the collected saliva are influenced by diet composition, smoking and alcohol consumption, as well as the presence of systemic and comorbid diseases, medications, radiotherapy, and the condition of the gums and teeth.
I thank the Reviewer for reviewing my work. I hope that the new version complies with the Reviewer’s suggestion, and may be again considered for publication in Biology.